# Effectiveness of Constraint-Induced Movement Therapy (CIMT) on Balance and Functional Mobility in the Stroke Population: A Systematic Review and Meta-Analysis

**DOI:** 10.3390/healthcare10030495

**Published:** 2022-03-08

**Authors:** Jaya Shanker Tedla, Kumar Gular, Ravi Shankar Reddy, Arthur de Sá Ferreira, Erika Carvalho Rodrigues, Venkata Nagaraj Kakaraparthi, Giles Gyer, Devika Rani Sangadala, Mohammed Qasheesh, Rakesh Krishna Kovela, Gopal Nambi

**Affiliations:** 1Department of Medical Rehabilitation Sciences, College of Applied Medical Sciences, King Khalid University, Abha 61413, Saudi Arabia; jtedla@kku.edu.sa (J.S.T.); kmeny@kku.edu.sa (K.G.); vnraj@kku.edu.sa (V.N.K.); drani@kku.edu.sa (D.R.S.); 2Postgraduate Program in Rehabilitation Science, University Center Augusto Motta UNISUAM, Rio de Janeiro 21032-060, Brazil; asferreira@unisuam.edu.br (A.d.S.F.); erikacrodrigues@gmail.com (E.C.R.); 3The London College of Osteopathic Medicine, London NW1 6QH, UK; info@osteon.co.uk; 4Department of Medical Rehabilitation Sciences, College of Applied Medical Sciences, Jazan University, Jazan 45142, Saudi Arabia; mqasheesh@jazanu.edu.sa; 5Department of Neuro Physiotherapy, Ravi Nair Physiotherapy College, Datta Meghe Institute of Medical Sciences, Sawangi (Meghe), Wardha 442001, Maharastra, India; rakeshkrishna.pt@gmail.com; 6Gopal Nambi, Department of Health and Rehabilitation Sciences, College of Applied Medical Sciences, Prince Sattam Bin Abdulaziz University, Al-Kharj 16278, Saudi Arabia; physio_gopal@rediffmail.com

**Keywords:** constraint-induced movement therapy, stroke, balance, functional mobility

## Abstract

Constraint-induced movement therapy (CIMT) is one of the most popular treatments for enhancing upper and lower extremity motor activities and participation in patients following a stroke. However, the effect of CIMT on balance is unclear and needs further clarification. The aim of this research was to estimate the effect of CIMT on balance and functional mobility in patients after stroke. After reviewing 161 studies from search engines including Google Scholar, EBSCO, PubMed, PEDro, Science Direct, Scopus, and Web of Science, we included eight randomized controlled trials (RCT) in this study. The methodological quality of the included RCTs was verified using PEDro scoring. This systematic review showed positive effects of CIMT on balance in three studies and similar effects in five studies when compared to the control interventions such as neuro developmental treatment, modified forced-use therapy and conventional physical therapy. Furthermore, a meta-analysis indicated a statistically significant effect size by a standardized mean difference of 0.51 (P = 0.01), showing that the groups who received CIMT had improved more than the control groups. However, the meta-analysis results for functional mobility were statistically insignificant, with an effect size of −4.18 (P = 0.16), indicating that the functional mobility improvements in the investigated groups were not greater than the control group. This study’s findings demonstrated the superior effects of CIMT on balance; however, the effect size analysis of functional mobility was statistically insignificant. These findings indicate that CIMT interventions can improve balance-related motor function better than neuro developmental treatment, modified forced-use therapy and conventional physical therapy in patients after a stroke.

## 1. Introduction

The second most common cause of death and disability worldwide was stroke [1], with more than 116 years of healthy life lost worldwide each year due to deaths and disability related to strokes [2]. Many advanced rehabilitation methods to treat patients after a stroke, including robotic-assisted technology [3,4,5], transcranial brain stimulation [2,5,6], virtual reality techniques [7], and game-based rehabilitation [8], have emerged in recent decades. Along with these advanced rehabilitation methods, the traditional approaches of neurodevelopmental treatment [9], proprioceptive neuromuscular facilitation [9,10,11], constraint-induced movement therapy (CIMT) [12], and task-oriented training [13] continue to be popular and used for the rehabilitation of patients after stroke to improve strength, balance, gait, function, and quality of life.

Among the traditional approaches, CIMT was first developed for the upper extremity and consisted of constraining the unaffected upper extremity to improve the function of the paralyzed upper extremity [14,15]. Researchers have since further utilized CIMT for the lower extremity and trunk to improve motor function [16,17,18,19,20]. Despite being designed to improve upper extremity function, many researchers have surprisingly noted improvements in balance as well [21,22,23].

Balance is the ability to use muscular forces to control the center of gravity both within and outside of the base of support [24]. Balance is one of the core determinants of independent gait [25] and quality of life [26] in the stroke population. During CIMT, when the upper extremity, trunk, or lower extremity is constrained, the patient is required to perform specific functional tasks prescribed by the therapist without the aid of the unaffected extremity. The patient thus must move the affected side, causing a shift in the center of gravity on the base of support that indirectly improves the central feedforward mechanisms to the muscular systems controlling the body and enhances balance [22].

There are many methods of objectively measuring balance in patients after stroke. Researchers have measured static, dynamic, and functional mobility components of balance among patient’s post-stroke. For example, the static element of balance has been measured by the center of pressure [27], center of mass [28], and symmetrical weight-bearing [29]. The dynamic component of balance has been measured using reach distances [30] and the Berg Balance Scale (BBS) [31], while functional mobility components have been measured by Dynamic Gait Index [31] and Timed Up and Go Test (TUG) [31].

The previous systematic review focused on finding the effects of lower extremity CIMT on balance and functional mobility have provided positive effects in their systematic review; however, a meta-analysis could not show the significant effect size [17] and recommended to conduct a future meta-analysis including more studies. Hence, CIMT influence on balance is unclear, effect of upper extremity restraint on balance has not been fully explored, and the existence of ambiguity in this treatment technique requires further investigation. Therefore, the aim of the current study was to evaluate the influence of and upper extremity and lower extremity CIMT on balance and functional mobility in patients after a stroke.

## 2. Methodology

### 2.1. Selection Criteria

Studies with an RCT design on patients above 18 years of age, with first-time or recurrent stroke, comparing either upper or lower extremity CIMT with traditional physical therapy, neurodevelopmental treatment, or proprioceptive neuromuscular facilitation, were enrolled in this study. The chief measurements of interest were balance (static, dynamic, and functional) and functional mobility. Studies published in languages other than English, studies incorporating the same treatment principles in both the groups, single session studies, and research designs except randomized controlled trials were excluded.

### 2.2. Search and Study Selection Process

RCTs available in the English language between January 2000 to December 2020 were searched using databases such as EBSCO, PubMed, PEDro, Science Direct, Scopus, and Web of Science. We used the following MeSH key terms to identify research articles: stroke, subacute, acute, chronic stroke, cerebrovascular accidents, hemiparesis, hemiplegia, constraint-induced movement therapy, balance, functional mobility, and lower extremity function. The details of the search strategy were mentioned in Table 1. Two independent reviewers (RR and VN) performed the online and offline search, grey literature search, and study selection process.

In the preliminary phase of study selection, the relevant articles were extracted by a reviewer (RS) based on recognizable titles and references. The full texts of the relevant articles were then reviewed by two authors (GK and DR) for their suitability for inclusion in the review. A third author (JT) was contacted to solve any differences of opinion arising between the authors regarding article inclusion. In the case of missing data, the study’s authors were contacted through email with a request to provide the information.

### 2.3. Quality Evaluation of Involved RCTs

The quality of the involved RCTs was analyzed by Physiotherapy Evidence Database (PEDro) scoring system. The third reviewer was consulted to resolve any discrepancies between the two independent reviewers regarding the designation of the scores for included studies. The studies were classified as poor, fair, good, and excellent based on the PEDro scoring system; poor if the grades are less than four, fair if the scores were between four to five, good if the scores were between six and eight, and excellent if the score were between nine and eleven. Two anonymous reviewers analyzed the methodological quality of the involved RCTs. Along with the methodological quality assessment, we also evaluated risk of bias assessment.

### 2.4. Data Analysis

To perform the meta-analysis, Review Manager (non-Cochrane mode) software was used. If an outcome was assessed through multiple scales, a pooled standardized mean difference (SMD) was implemented to calculate the effect size. If an outcome was assessed using a single scale, a pooled mean difference (MD) was utilized. I2 statistics were used to evaluate the heterogeneity among the studies for each outcome. An I2 value of more than 50% indicated considerable heterogeneity among the studies. Based on the I2 heterogeneity test results and the clinical heterogeneity, either fixed- or random-effects models were employed.

## 3. Results

### 3.1. Search Results

Eight studies were included after reviewing 161 non-duplicate research articles. Further elements of the search and process of selection are described in Figure 1. Across the eight studies, 208 subjects participated in the trials (127 males and 81 females).

### 3.2. Quality Assessment of the Involved RCTs

Among the included studies, one study obtained a PEDro score of 4 [32], three studies obtained a score of 5 [33,34,35,36], three studies obtained a score of 6 [20,21,22], and one study obtained a score of 8 [37]. Based on the quality assessment rating, four studies were rated as fair [32,34,35,36] and four studies were rated as good [18,21,23,33]. Details of the PEDro scores attained by the involved RCTs are provided in Table 2. The findings of risk of bias assessment showed that, overall, there is risk of concealed allocation, attrition bias, blinding of participants and therapist. The details of risk of bias assessment are given in Figure 2.

### 3.3. Descriptions of Included RCTs

The mean age of the population in the included studies ranged from 52.46 years to 61.58 years. The included subjects had a chronic stroke [22,23,32,35,36] in all but two studies [21,34]. Out of the eight randomized controlled trials, two assessed upper extremity CIMT [22,33], and the remaining six investigated lower extremity CIMT [21,23,32,35,36]. The control group interventions primarily included conventional physiotherapy concepts such as muscle strengthening, facilitation, activity training, balance, and gait training; however, in two studies, the control intervention involved concept-based interventions such as NDT [21,35]. The treatment duration per session ranged from 30 min to 120 min, though 30 min per session was the most common duration utilized in the studies. The total number of sessions ranged from 6 to 20, and the number of treatment sittings for each week alternated from 1 to 5. The descriptions of the involved RCTs were represented in Table 3.

### 3.4. Outcome Measures

The outcome measures utilized to assess balance were the BBS, limits of stability (LOS) toward the affected side and toward the unaffected side, center of mass displacement, center of pressure translation in medial to lateral and anterior to posterior directions, Functional Reach Test (FRT), modified-FRT, and Trunk Impairment Scale (TIS). The TUG test was the most common outcome measure employed to measure functional mobility. Of the eight included randomized controlled trials, three studies showed superior improvement in the investigated groups when compared to the control group [33], and the remaining studies showed equal effects [22,23,32,34,36] in both groups, with none of the studies showing less or negative effects in the CIMT intervention group in comparison to the control intervention group. Additional details regarding the study characteristics can be found in Table 3.

### 3.5. Quantitative Synthesis

The quantitative synthesis comprised a separate meta-analysis of balance and functional mobility. In the balance meta-analysis, the BBS, LOS, TIS, and FRT outcome measures were included, so the SMD was used to assess the effect size. As the TUG test was the only outcome measure used to assess functional mobility, the MD was chosen for calculating its effect size. The balance meta-analysis showed an effect size with SMD of 0.51 (95% CI [0.12–0.91], P = 0.01), indicating that the experimental group CIMT intervention was superior in improving balance among patients after stroke than the control group interventions. The heterogeneity of the included studies was I2 = 55% (P = 0.03), indicating moderate variability among the included studies. The subgroup analysis had interesting findings for upper and lower extremity CIMT. Upper extremity CIMT had the effect size with SMD of 0.55 (95% CI [0.18–0.92], P = 0.04). The heterogeneity of the included studies was I2 = 0% (P = 0.57), indicating variability among the included studies. Lower extremity CIMT had the effect size with SMD of 0.56 (95% CI [−0.15 to 1.26], P = 0.12). However, the heterogeneity of the included studies was I2 = 71% (P = 0.008) which shows a high level of variability among the included studies (Figure 3).

The meta-analysis of functional mobility for the TUG test showed a statistically insignificant MD, indicating that CIMT was not superior to the control group in improving functional mobility. Moreover, the subgroup analysis of both upper and lower extremity CIMT was also statically insignificant. The statistical details of the meta-analysis of functional mobility are shown in Figure 4.

## 4. Discussion

Previous reviews of CIMT involved many impairment measures, such as range of motion and spasticity, and activity measures, such as gait and upper and lower extremity function, though many did not consider the crucial components of balance and functional mobility [17,37,38,39,40]. The current systematic review with meta-analysis of RCTs is thus the first to focus on the effects of lower and upper extremity CIMT on balance in the stroke population.

The results of the systematic review indicated that the CIMT has either positive effects or equal effects compared to the controlled interventions. There are many factors which may influence these results, such as type of interventions in the control group, type of constraints, duration of immobilization, duration of intervention, and variability in the balance outcome measures. While considering the duration of the interventions, the Zhu et al. [34] study found significant effects of CIMT on center of mass displacements. However, the unequal distribution of duration of intervention favoring the CIMT group might be the reason for the above positive effects. With regard to the intervention, there is variability in the control groups; the majority of them used conventional physical therapy [32,34,36], some studies used neurodevelopmental treatment [21,35], some authors used modified force use of upper extremity [22], and others used treadmill and overground gait training [23,33]. This variability of the interventions might provide the reasons for inconsistent improvements found in the control groups. Further, this factor would have favored the improvements in the experimental group as whole. As per the core principles of CIMT, behavioral modification, repetitions, transfer package and constraint are important. In the current systematic review, two authors [32,34] used augmentation by shoe inserts and encouragement of weight bearing on the affected side, and others used restraint of unaffected limb [21,22,23,33,34,35], to maximize the participation of the affected limb. However, both restraint and augmentation showed the positive effects on improving the balance of the patients.

Our findings were comparable to the conclusions of the study by Abdullahi et al. [17], which focused exclusively on the effects of lower extremity CIMT on the stroke population, including multiple lower extremity functional components such as gait, lower extremity motor function, balance, functional mobility, and quality of life. Their effect size from the meta-analysis was 0.62 (95% CI [−0.54 to 1.78]) for balance and −0.53 (95% CI [−3.61 to 2.55]) for functional mobility, showing a statistically insignificant effect of lower extremity CIMT on both balance and functional mobility. In our study, the effect size calculated from the meta-analysis of balance was 0.51 (95% CI [0.1–0.91]), which indicated a statistically significant effect of CIMT on balance; however, when we examined functional mobility in a meta-analysis, the effect size was −2.73 (95% CI [−8.59 to 3.13]), indicating a statistically insignificant effect of CIMT on functional mobility. Nevertheless, the significant effect size in the meta-analysis on balance supports the positive effects of CIMT on balance, which may be due to the inclusion of studies focusing on upper extremity CIMT. In addition, in Abdullahi et al. [17], three research works were incorporated in the meta-analysis of balance, including the BBS and FRT outcomes. In contrast, in our study, seven research works were incorporated in the meta-analysis of balance, and their outcome measures were the BBS, FRT, LOS, and TIS.

The subgroup analysis of the effect of upper extremity CIMT on balance was statistically significant, with an effect size of 0.55 (95% CI [0.18–0.92]) and homogeneity. In contrast, the effect of lower extremity CIMT on balance was statistically insignificant, with the effect size of 0.56 (95% CI [−0.15 to 1.26]) and heterogeneity of I2 = 71%. This may be due to variability in the lower extremity CIMT methodology in terms of the type of constraint, duration of treatment, and type of control group intervention. Moreover, the upper extremity CIMT has a substantial influence on trunk control in practice; the constraint of the normal upper extremity forces the affected upper extremity to move. Without adequate upper extremity control, patients undergoing CIMT might use their trunk to aid in the upper extremity movement, thus leading to a greater shift in the center of gravity on the base of support and better improvement in the balance outcome measures [22,41,42].

In our research, we establish an important influence of CIMT on balance but not on functional mobility as measured by the TUG test. The outcome measures assessing balance primarily focus on static and dynamic components of balance, while the TUG test involves not only static and dynamic components but also walking and turning; this might have contributed to the insignificant effect size [43]. In the current literature, there are only a few studies addressing functional mobility; in the future, a larger number of studies involving this outcome measure may change the significance.

The meta-analysis of post-intervention balance scores showed a significant effect size of 0.51 (95% CI [0.12–0.91]). However, the heterogeneity among the included studies was I2 = 55%, indicating variability in the studies in terms of sample size, methodological quality, duration of the stroke, duration of intervention, type of outcome measure and constraint. Moreover, the risk of bias assessment had revealed some additional factors which might have influenced the study results, such as bias in the patient allotment to the groups, assessment, and handling of incomplete data. The subgroup analysis based on the duration of immobilization and chronicity of stroke did not cause any noticeable change in the results of the meta-analysis.

Future studies must consider aforementioned factors to maintain a high research quality. Therapists may hypothesize that upper extremity CIMT will not have a significant influence on balance, but in our meta-analysis, we saw the surprising result that the effect size became insignificant when we excluded the upper extremity studies from the sensitivity analysis, indicating the influence of the upper extremity findings on the overall effect size.

In the present study, a small number of randomized controlled trials satisfied the selection criteria, causing us to include moderate-quality studies in our meta-analysis. Both lower and upper extremity CIMT were included; in the future, if enough randomized controlled trials are available, researchers should assess the effects of lower and upper extremity CIMT independently. Future RCTs should focus on high methodological quality, incorporating similar intervention protocols among the groups with and without restraints, and observing the effects of CIMT at follow-up. Trunk CIMT was commonly assessed in the literature though not included in this review because balance was not considered as an outcome measure; trunk CIMT should be further evaluated in future studies.

## 5. Conclusions

This systematic review revealed positive effects of CIMT on balance in three studies and equal effects in five studies, compared to the control group interventions such as conventional physical therapy, NDT, gait training and forced-use therapy. A meta-analysis demonstrated a positive effect size of 0.51 (P = 0.01), showing that balance improved more with the experimental group CIMT intervention than control group interventions. Moreover, we observed that upper extremity CIMT interventions elicited greater improvements in balance than lower extremity CIMT. However, the meta-analysis of functional mobility had a statistically insignificant effect size of −4.18, indicating that functional mobility improvements in the investigated groups were not superior to the control group.

## Figures and Tables

**Figure 1 healthcare-10-00495-f001:**
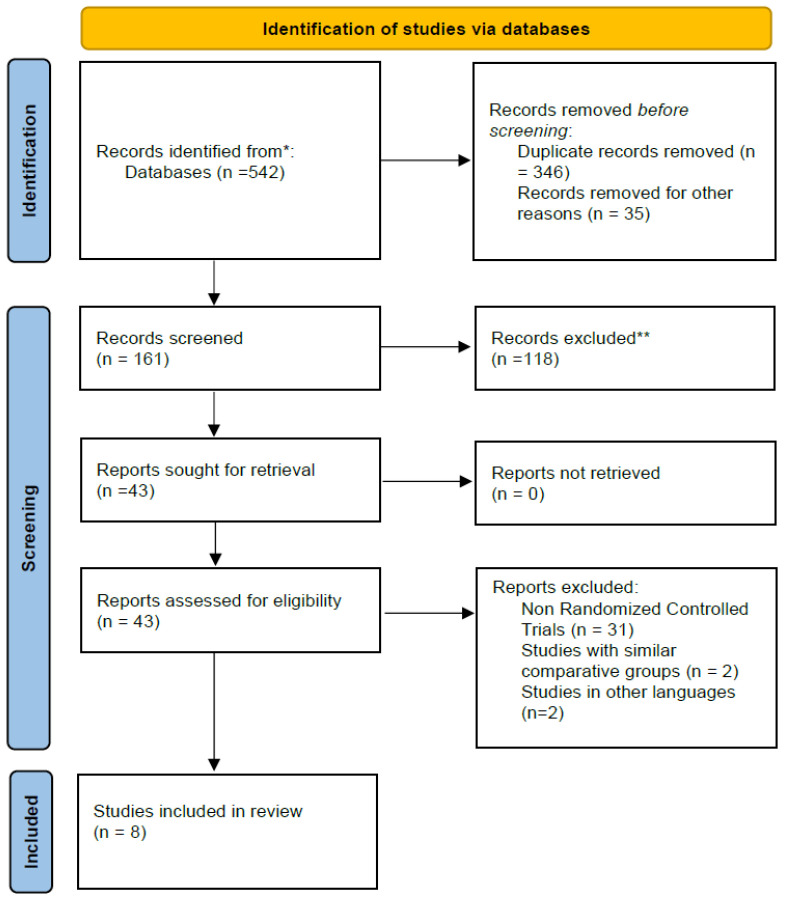
Flowchart showing the process of data collection and analysis.

**Figure 2 healthcare-10-00495-f002:**
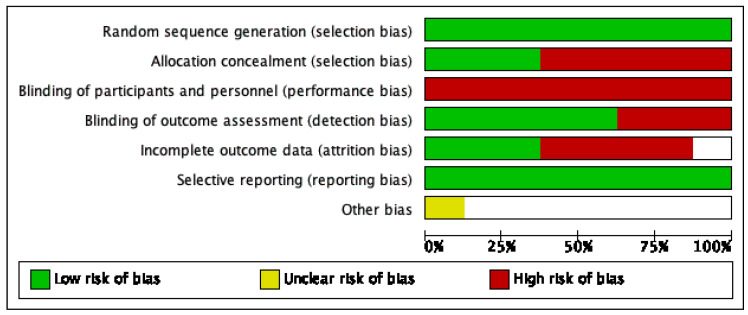
Risk of bias assessment of the included studies.

**Figure 3 healthcare-10-00495-f003:**
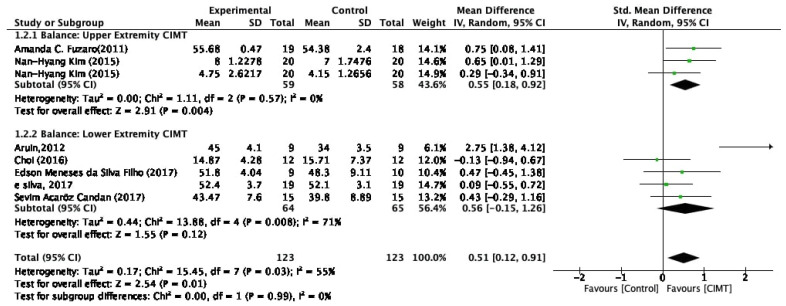
Meta-analysis results on the effect of CIMT on balance among stroke population.

**Figure 4 healthcare-10-00495-f004:**
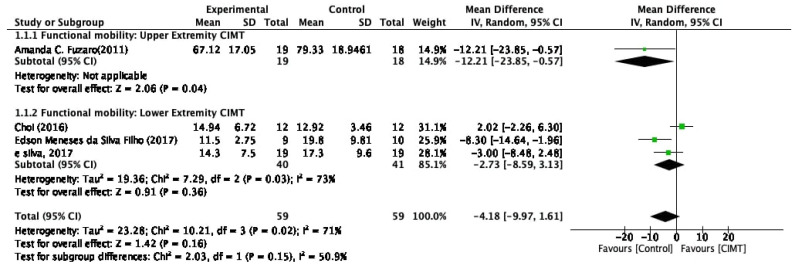
Meta-analysis results on the effect of CIMT on functional mobility among stroke population.

**Table 1 healthcare-10-00495-t001:** Search strategy used in the study.

Databases (Number of Articles)	PICO Format Search with Bullion Key Words (and)(or)			
	Patient	Intervention	Comparison	Outcome
EBSCO (95) PubMed (96) PEDro (47) Science Direct (154) Scopus (81) Web of Science (69)	Stroke OR Hemiplegia OR Hemiparesis OR Cerebrovascular accident	CIMT OR Constraint-induced movement therapy OR Restricted Limb/Extremity OR	Neuro Developmental Treatment OR NDT OR Proprioceptive Neuromuscular Facilitation OR PNF OR Conventional Physical Therapy OR CPT OR Physiotherapy OR Exercise OR Forced Arm Use Or Traditional Rehabilitation OR Standard Physical Therapy	Balance OR Functional Mobility OR Center Of Gravity OR Base of Support OR Center of Pressure

**Table 2 healthcare-10-00495-t002:** Quality evaluation of involved RCT’s by PEDro scale.

S.No	Author/Year	Eligibility Criteria	Random Allocation	Concealed Allocation	Baseline Comparability	Blinding of Participants	Blinding of Therapist	Blinding of Assessor	Adequate Follow up (>85%)	Intention to Treat	Between Group Comparison	Point Estimates and Variability	Pedro Score [10]
1	Aruin AS et al., 2012 [32].	Yes	Yes	No	Yes	No	No	No	No	No	Yes	Yes	4
2	Fuzaro AC et al., 2012 [22].	Yes	Yes	Yes	Yes	No	No	Yes	No	No	Yes	Yes	6
3	Kim NH et al., 2015 [33].	No	Yes	No	Yes	No	No	No	Yes	No	Yes	Yes	5
4	Zhu Y et al., 2016 [34].	Yes	Yes	No	Yes	No	No	Yes	No	No	Yes	Yes	5
5	e Silva EMG de S et al., 2017 [23].	Yes	Yes	Yes	Yes	No	No	Yes	Yes	Yes	Yes	Yes	8
6	Candan SA et al., 2017 [35].	Yes	Yes	No	Yes	No	No	Yes	Yes	No	Yes	Yes	6
7	Choi HS et al., 2017 [21].	Yes	Yes	Yes	Yes	No	No	No	Yes	No	Yes	Yes	6
8	Silva EM et al., 2017 [36].	Yes	Yes	No	Yes	No	No	Yes	No	No	Yes	Yes	5

**Table 3 healthcare-10-00495-t003:** Characteristics of studies on constraint-induced movement therapy (CIMT).

S.No	Author/Year	Age	Chronicity	Intervention		Duration	Outcome Measures	Inferences
				Experimental	Control			
1	Aruin AS et al., 2012 [32].	57.7 ± 311.9	6.7 ± 3.9 Years	The management focused on accelerating muscle strength, balance, and symmetrical weight bearing by forcing body shift towards affected lower limb by shoe insert	Conventional physical therapy	60 min per session, 1 session per week, six sessions—in total 6 h	SWB, gait speed (m/s), BBS, FM for lower limb	Post- and follow-up retention were observed for symmetrical weight bearing, gait speed, and BBS in experimental group.
2	Fuzaro AC et al., 2012 [22].	52.46 ± 14.29	Experimental 19.5 ± 20.8 Months Control 30.8 ± 31.8 Months	Bimanual activities in special tasks with paretic upper limb as a main conductor with immobilization of non-paretic upper limb	Modified forced-use therapy	50 min per single session, five sessions each week for 4 weeks	SIS, BBS, FM, T10, and TUG	Both m-CIMT and m-FUT showed equal improvements for balance, functional mobility, motor functions and quality of life at post- and follow-up sessions
3	Kim NH et al., 2015 [33].	54.75 ± 4.9	Experimental 24.1 ± 10.7 Months Control 30.8 ± 11.0 Months	Overground gait training with non-paretic upper-limb constraint	Overground gait training without non-paretic upper- limb constraint	30 min per session, three sessions each week for 4 weeks adding to central nervous system developmental treatment for 60 min per session, five times each week, over the same 4 weeks	TIS (static, dynamic and coordination) LOS TAS and LOS TUS (cm)	Experimental group showed significant improvements for TIS and LOS
4	Zhu Y et al., 2016 [34].	58.71 ± 6.02	Experimental 3.90 ± 0.83 Months Control 3.72 ± 0.78 Months	Gait training consists of 2 h of sit-to-stand transfers, indoor walking, climbing up and down stairs, balance training and one-leg weight training with more repetitions in addition to this standard comprehensive rehabilitation for 45 min	Conventional physical therapy	45 min per session, five sessions per week for four weeks	COM displacements gait speed (m/s), step width (m), step length paretic and no-paretic side (m). Paretic and non-paretic swing time (%GC)	m-CIMT gait training improved both COM displacements and spatial-temporal gait parameters
5	e Silva EMG de S et al., 2017 [23].	57.75 ± 3.75	Experimental 3.5 ± 1.73 Months Control 3.7 ± 1.4 Months	Treadmill training with ankle weight on normal lower limb	Treadmill training without restraint	Nine training sessions, 30 min per session, 2 consecutive weeks	BBS, turn speed (m/s), TUG, stride time (s), stride length (s), symmetry ratio, and stride width (m)	Spatial-temporal gait parameters, balance and functional mobility improved in both groups equally.
6	Candan SA et al., 2017 [35].	56.4 ± 13.45	Experimental 6.8 ± 2.7 Months Control 6.6 ± 3.1 Months	m-CIMT includes intensive practice, restraint of non-paretic lower limb and transfer package	NDT program	120 min per session, five sessions per week, for 2 weeks	BBS, FAC, gait speed, cadence (steps/min), step length ratio and postural symmetry	Balance, functional ambulation, gait speed, and step length ratio improved significantly in m-CIMT group when compared to NDT group
7	Choi HS et al., 2017 [21].	61.58 ± 5.83	Experimental 13.5 ± 5.5 Months Control 14.2 ± 4.8 Months	Game-based CIMT group undertook game-based CIMT and conventional physical therapy	Conventional physical therapy including NDT	Game-based CIMT for 30 min in a session, for three sessions a week for 4 weeks. All subjects, comprising those in the control group, underwent conventional physical therapy for 60 min a session, five sessions a week for 4 weeks	COP displacements medial-lateral and anterior-posterior (cm), sway area (cm^2^), sway mean velocity (cm/s), SWB, FRT (cm), m-FRT (cm), and TUG (s)	Game-based CIMT showed significant improvements in static balance, symmetrical weight bearing, and medial-lateral shift compared to control group
8	Silva EM et al., 2017 [36].	55.63 ± 4.93	Experimental 13.7 ± 3.4 Months Control 29.2 ± 10.1 Months	Encouraging paretic limb to perform specific activities with constraint of non- paretic upper limb	Conventional physical therapy without constraining non-paretic upper limb	30 min per each session, three sessions per week for 4 weeks	BBS, TUG, stairs, and gait speed	Post-intervention improvement was observed in experimental group for gait speed, BBS, TUG, and stairs up and down without any significant difference between the groups

**Notes:** a. BBS: Berg Balance Scale, b. (m/s): (meters/second), c. (m): meters, d. (cm): centimeters, e. SWB: symmetrical weight bearing, f. FM: Fugl-Meyer, g. (%GC): percentage of gait cycle, h. COP: center of pressure, i. TIS: trunk impairment scale, j. TUG: Timed Up and Go test, k. FAC: functional ambulation category, l. SIS: stroke impact scale, m. m-FRT: modified Functional Reach Test, n. FRT: Functional Reach Test, o. (cm/s): centimeters/second, p. LOS: limits of stability, q. (TAS): towards affected side, r. (TUS): towards unaffected side, s. (COM): center of mass, t. m-FUT: modified forced-use therapy.

## Data Availability

Data will be available on request with the primary author jtedla@kku.edu.sa.

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
