# Peer review of "Effectiveness of Constraint-Induced Movement Therapy (CIMT) on Balance and Functional Mobility in the Stroke Population: A Systematic Review and Meta-Analysis"

_healthcare, 2022, doi:10.3390/healthcare10030495_

Round 1

Reviewer 1 Report

This study addresses a very interesting therapy for post-stroke rehabilitation, focusing on its effects on balance. In general, the writing is good and the references are very up-to-date. However, it is necessary to pay attention to some methodological aspects in order to deal with this systematic review. In addition, it would be necessary to enrich the discussion, discussing between different types of balance and contrasting the results in a more analytical way, launching hypotheses about them, supported by bibliographic references. Here are some observations that should be taken into account:

  • Lines 28 and 36: It would be necessary to specify what the control interventions consist of.

  • Line 38: It is recommended not to use acronyms as keywords and, whenever possible, to use MeSH terms.

  • Lines 66-75: It is assumed that the selected studies are not yet known in the Introduction and should not be referred to directly. I consider it a good idea to introduce balance measurement tools, justifying their validity through studies or reviews focused on evaluation, but in a generic way and not making direct reference to the selected studies.

  • Lines 76-78: In addition to the aim of the review, a justification of the present study is necessary. Previous reviews on the subject that are referenced in the discussion could be mentioned here and specify what this review brings new. For example, reference 17 specificallyaddresses balance as an outcome. It is recommended to also include an existing Cochrane review on this topic.

  • Lines 80-86: It is necessary to include exclusion criteria and whether the review followed the PRISMA 2020 principles, including the PICO model and PROSPERO code.

  • Line 89: Google Scholar more than a database, is considered gray literature. Please specify databases, rather than search engines.

  • Lines91-92: Was the term constraint-induced movement therapy not included in the search? It is essential to include a table with the specific search strategies used in each of the databases.

  • Line 102: In addition to methodological quality, it is recommended to assess the risk of bias using the Cochrane tool.

  • Line 125: It is necessary to use a flowchart model based on PRISMA, which specifies the number of studies obtained in each of the databases, the number of duplicates excluded, and the number of studies excluded for each cause of exclusion. In addition, the number of studies or gray literature located manually.

  • Line 144 (Table 2): In colum 1, Including references, along with the name of the authors and year of publication, would facilitate the location of the studies; in colum 3, It is recommended to include the mean number of months post-stroke, since the term chronicity can cover a wide range and this can greatly affect the results obtained; in colum 5, Regarding the control group, it would be necessary to pool the results when CIMT was compared with the same intervention but without restricting that side or when it was compared with conventional or traditional rehabilitation; In the intervention group of the study by Zhu et al. a two-hour intervention was performed. Could the intensity of treatment influence this group, apart from the type of intervention? It would be necessary to discuss it.

  • Discussion an Conclusion: It is necessary to take into account the assessments made in Table 2 in the Discussion and Conclusion section.It is necessary to specify that the control groups of the meta-analysis carry out the same approach as the intervention group, but without restricting that side.

Author Response

Response to reviewer comments

Thank you so much for your valuable time to review and for considering our study. The suggestions offered by the reviewers were immensely helpful. We carefully considered changes in the manuscript as per your suggestions. We modified and highlighted the changes in the manuscript. The highlighted changes are in yellow color. Here is a point-by-point explanation of reviewer comments. Thank you once again.

Reviewer 1

Sl.no

Queries

Response to queries

Line Numbers

1

This study addresses a very interesting therapy for post-stroke rehabilitation, focusing on its effects on balance. In general, the writing is good and the references are very up-to-date. However, it is necessary to pay attention to some methodological aspects in order to deal with this systematic review. In addition, it would be necessary to enrich the discussion, discussing between different types of balance and contrasting the results in a more analytical way, launching hypotheses about them, supported by bibliographic references. Here are some observations that should be taken into account:

Thank you so much for your valuable time to review and for considering our study. The suggestions offered by the reviewers were immensely helpful. We carefully considered changes in the manuscript as per your suggestions. We modified and highlighted the changes in the manuscript. The highlighted changes are in yellow color. Here is a point-by-point explanation of reviewer comments. Thank you once again

N/A

2

Lines 28 and 36: It would be necessary to specify what the control interventions consist of.

As per the suggestions we specified the control interventions in the abstract

27 – 30 and 36 - 38

3

Line 38: It is recommended not to use acronyms as keywords and, whenever possible, to use MeSH terms.

We modified our keyword as per the suggestion

40

4

Lines 66-75: It is assumed that the selected studies are not yet known in the Introduction and should not be referred to directly. I consider it a good idea to introduce balance measurement tools, justifying their validity through studies or reviews focused on evaluation, but in a generic way and not making direct reference to the selected studies.

We changed the reference numbers as per the suggestion and modified the introduction.

75 - 83

5

Lines 76-78: In addition to the aim of the review, a justification of the present study is necessary. Previous reviews on the subject that are referenced in the discussion could be mentioned here and specify what this review brings new. For example, reference 17 specifically addresses balance as an outcome. It is recommended to also include an existing Cochrane review on this topic.

As per the recommendation we provided the justification for the present study and mentioned about the details of previous review.

75 - 83

6

Lines 80-86: It is necessary to include exclusion criteria and whether the review followed the PRISMA 2020 principles, including the PICO model and PROSPERO code.

As per the suggestion of the reviewers the exclusion criteria were mentioned and the study followed PRISMA 2020 principles, including the PICO model and PROSPERO code.

90 - 93

7

Line 89: Google Scholar more than a database, is considered gray literature. Please specify databases, rather than search engines.

We modified databases according to your recommendations.

96 - 97

8

Lines91-92: Was the term constraint-induced movement therapy not included in the search? It is essential to include a table with the specific search strategies used in each of the databases.

We are ashamed of this mistake and thank you for pointing it out. The term constraint-induced movement therapy is there in the literature search and search strategy table also added now as per your suggestion.

99 – 100

103 – 104 (Table 1)

9

Line 102: In addition to methodological quality, it is recommended to assess the risk of bias using the Cochrane tool.

As per the comment. We now included the Risk of bias assessment and related information is mentioned in methodology, results and discussion.

120 – 121

143 – 146

Figure 2 (147 -148).

308 - 312

10

Line 125: It is necessary to use a flowchart model based on PRISMA, which specifies the number of studies obtained in each of the databases, the number of duplicates excluded, and the number of studies excluded for each cause of exclusion. In addition, the number of studies or gray literature located manually.

The flowchart is now modified as per the PRISMA guidelines

Figure 1 (135 – 137)

11

Line 144 (Table 2): In colum 1, Including references, along with the name of the authors and year of publication, would facilitate the location of the studies; in colum 3, It is recommended to include the mean number of months post-stroke, since the term chronicity can cover a wide range and this can greatly affect the results obtained; in colum 5, Regarding the control group, it would be necessary to pool the results when CIMT was compared with the same intervention but without restricting that side or when it was compared with conventional or traditional rehabilitation; In the intervention group of the study by Zhu et al. a two-hour intervention was performed. Could the intensity of treatment influence this group, apart from the type of intervention? It would be necessary to discuss it.

As per the suggestion the references were added in column one, and mentioned mean months of the chronicity in the column three. Clarified about the control group interventions in the column five.

We discussed about impact of Zhu et al. study also in the discussion.

Table 2.

Column 1, 3 and 5.

252 – 254.

Discussion and Conclusion: It is necessary to take into account the assessments made in Table 2 in the Discussion and Conclusion section. It is necessary to specify that the control groups of the meta-analysis carry out the same approach as the intervention group, but without restricting that side.

As per your recommendation we revised the conclusion and discussion.

252 – 266

325 - 329

330 -339

Reviewer 2 Report

This systematic review and meta-analysis report the effectiveness of CIMT on balance and functional mobility amongst people affected by stroke. Although the review part is acceptable, conclusions of the meta-analysis are not rigorous due to the limited number of studies included and their wider heterogeneity.

Author Response

Response to Reviewer comments

Reviewer 2

Sl.no

Queries

Response to queries

Line Numbers

1

This systematic review and meta-analysis report the effectiveness of CIMT on balance and functional mobility amongst people affected by stroke. Although the review part is acceptable, conclusions of the meta-analysis are not rigorous due to the limited number of studies included and their wider heterogeneity.

Thank you so much for your valuable time to review and for considering our study. The suggestions offered by you and other reviewers were immensely helpful. We carefully considered changes in the manuscript as per your suggestions. We carefully considered your comments and reported the reasons for heterogeneity and its impact in detail in the discussion.

252 – 266

307 – 312

323 – 325

330 -339.

Reviewer 3 Report

Thank you for the opportunity to review this interesting manuscript.

The point was: The effect of CIMT on balance has not been fully explored, and the existence of ambiguity in this treatment technique requires further investigation. Therefore, the aim of the current study was to evaluate the influence of CIMT on balance and functional mobility in patients after a stroke.

The study was planned correctly and in accordance with the accepted standards.

However, my concern is the selection of 8 articles.

  1. Aruin AS et al., 2012

To evaluate the effectiveness of the Compelled Body Weight Shift (CBWS) therapy involved a forced shift of body weight towards a person’s affected side by means of a shoe insert that established a lift of the non-affected lower extremity. Eighteen individuals with chronic, unilateral stroke (mean age 57.7 ± 11.9 years, with a range of 35–75 years, mean time since stroke 6.7±3.9 years with a range of 1.1–14.1 years) Both groups underwent a battery of identical tests (Fugl-Meyer assessment, Berg Balance Scale, weight bearing, and gait velocity) before the start of the rehabilitation intervention, following its completion, and three months after the end of therapy.

  1. Fuzaro AC et al., 2012 - To evaluate the effect of Modified FUT (mFUT) and Modified CIMT (mCIMT) on the gait and balance during four weeks of treatment and 3 months follow-up.

This study included thirty-seven hemiparetic post-stroke subjects that were randomly allocated into two groups based on the treatment protocol. The non-paretic UL was immobilized for a period of 23 hours per day, five days a week. Participants were evaluated at Baseline, 1st, 2nd, 3rd and 4th weeks, and three months after randomization. For the evaluation used The Stroke Impact Scale (SIS), Berg Balance Scale (BBS) and Fugl-Meyer Motor Assessment (FM). Gait was analyzed by the 10-meter walk test (T10) and Timed Up & Go test (TUG). Both groups revealed a better health status (SIS), better balance, better use of lower limb (BBS and FM) and greater speed in gait (T10 and TUG), during the weeks of treatment and months of follow-up, compared to the baseline. The results show mFUT and mCIMT are effective in the rehabilitation of balance and gait.

  1. Kim NH et al., 2015 - The present study was undertaken to examine the effect of intensive gait training using a constrained induced movement therapy (CIMT) technique for the non-paretic upper extremity on the balance ability of stroke patients.

Homogeneity testing of the two groups revealed significant differences in the groups’ ages. The major limitation of the present study was the small number of study subjects, which makes it difficult to generalize our results in the context of proposing an intervention for improving the balance of stroke patients.

4. Zhu Y et al., 2016 - his study aimed to qualify the improvements of modified constraint-induced movement therapy (m-CIMT) on the lower limb of stroke patients via assessing the centre of mass (COM) displacement and the basic gait parameters. A total of 22 hemiplegic patients after stroke with first-time clinical cerebral infarction or haemorrhagic cerebrovascular accident were included in this study. The patients were randomly divided into m-CIMT group and the conventional therapy group (control group), and received corresponding training for five days/week for four weeks. The COM displacement and gait parameters were assessed by three-dimensional segmental kinematics method in pre-intervention and post- intervention therapy. After four weeks of m-CIMT, the COM displacement on sagittal plane of paretic leg during stance phase was increased (pre: 91.04 ± 4.39 cm, post: 92.38 ± 4.58 cm, p < 0.05) and swing range of frontal plane was remarkably decreased (pre: 10.15 ± 3.05 cm, post: 7.83 ± 1.90 cm, p < 0.001). Meantime, the normalised swing range of COM in m-CIMT was superior to that in control group. Moreover, the gait parameters, including velocity (0.27 m/s), step width (0.10 m), step length (0.22 m) and swing time percentage (29.80%), were significantly improved by post-interventions of m-CIMT (p < 0.05). The m-CIMT intervention improves the COM displacement in sagittal and frontal plane, as well as gait parameters. These suggest that m-CIMT intervention may be feasible and effective for the rehabilitation of hemiplegic gait. Implications for Rehabilitation Segmental kinematics method was used to estimate the displacement of the COM. m-CIMT interventions improved the COM displacement of patients after stroke. m-CIMT interventions improved the hemiplegic gait parameters.

  1. e Silva EMG de S et al., 2017 - A 40-day follow-up, single-blind randomized controlled trial was performed with 38 subacute stroke patients (mean of 4.5 months post-stroke). Participants were randomized into: treadmill training with load to restraint the non-paretic ankle (experimental group) or treadmill training without load (control group). Both groups performing daily training for two consecutive weeks (nine sessions) and performed home-based exercises during this period. As outcome measures, postural balance (Berg Balance Scale - BBS) and functional mobility (Timed Up and Go test - TUG and kinematic parameters of turning - Qualisys System of movement analysis) were obtained at baseline, mid-training, post-training and follow-up. Repeated-measures ANOVA showed improvements after training in postural balance (BBS: F = 39.39, P < .001) and functional mobility, showed by TUG (F = 18.33, P < .001) and by kinematic turning parameters (turn speed: F = 35.13, P < .001; stride length: F = 29.71, P < .001; stride time: F = 13.42, P < .001). All these improvements were observed in both groups and maintained in follow-up. These results suggest that two weeks of treadmill gait training associated to home-based exercises can be effective to improve postural balance and functional mobility in subacute stroke patients. However, the load addition was not a differential factor in intervention.
  2. Candan SA et al., 2017 - In this study, investigated the effects of L-CIMT including the transfer package to induce behavioral transformation in normal daily life of patients with lower limb paralysis. The L-CIMT including the transfer package was administered to three patients with chronic-phase stroke without any constraint on the healthy lower limb for 3.5 hours a day, 5 days a week for 3 weeks. As a result, standing balance and walking ability were improved immediately and within 6 months after the intervention, respectively. All three cases experienced increased daily opportunities for standing and walking. We believe that L-CIMT including the transfer package can bring both short- and long-term improvements in standing balance and walking ability. This can lead to an increase in the frequency of standing and walking in daily living, along with an expanded range of action in ADL and IADL in patients with chronic-phase stroke.
  3. Choi HS et al., 2017 - the aims of this work were to determine whether game-based constraint-induced movement therapy (CIMT) is effective at improving balance ability in patients with stroke, and to provide clinical knowledge of game-based training that allows application of CIMT to the lower extremities. Thirty-six patients with chronic stroke were randomly assigned to game-based CIMT (n = 12), general game-based training (n = 12), and conventional (n = 12) groups. All interventions were conducted 3 times a week for 4 weeks. The static balance control and weight-bearing symmetry were assessed, and the Functional Reach Test (FRT), modified Functional Reach Test (mFRT), and Timed Up and Go (TUG) test were performed to evaluate balance ability. All 3 groups showed significant improvement in anterior-posterior axis (AP-axis) distance, sway area, weight-bearing symmetry, FRT, mFRT, and TUG test after the intervention (P < 0.05). Post hoc analysis revealed significant differences in AP-axis, and sway area, weight-bearing symmetry of the game-based CIMT group compared with the other group (P < 0.05). Although the general game-based training and the game-based CIMT both improved on static and dynamic balance ability, game-based CIMT had a larger effect on static balance control, weight-bearing symmetry, and side-to-side weight shift.
  4. Silva EM et al., 2017 - The Constraint Induced Movement Therapy (CIMT) can assist in the recovery of patients with post cerebrovascular accident sequelae. The aim was to assess whether the modified CIMT interferes with the balance and functional mobility of individuals in the chronic phase post-CVA, conducted a randomized, blinded, clinical trial with 19 patients in the chronic phase post-CVA. Group 1, “no constraint,” was submitted only to the paretic upper limb (UL) specific training (shaping). Group 2, “with constraint,” was submitted to the paretic UL specific training (shaping) and non paretic UL constraint. The training was carried out 3 times a week for 4 consecutive weeks. The volunteers were evaluated before and immediately after the sessions with the Berg Balance scale (BBS), Timed “up & go” (TUG), evaluation of gait speed and going up and down stairs. Mann-Whitney test showed that the balance (BBS) showed significant improvement (p=0.014) in the group that used the constraint in the intra-group analysis. There was improvement in the gait speed (p=0.050) in the intergroups analysis. It was concluded that the modified CIMT influenced in the balance and gait speed of the Group submitted to the paretic UL specific training and constraint in the non-paretic UL.

In addition:

I suggest that each article should be cross-referenced in Tables 1 and 2. This will make it easier for the reader to find a suitable artikle in the references.

 The presented study presents a differentiated CIMT methodology in terms of the type of restriction, duration of treatment and type of intervention in the control group.

One study used additional NDT therapy? Did this intervention not affect the results of this study?

Only one study used an additional factor, a shoe insert

One study looked at gait with restricted upper limb

One study does not look at people after stroke.

Only 3 patients participated in one study.

Moreover, some of the subjects were in the subacute phase, others in the chronic phase.

I find the manuscript interesting in general, but more as an overview of selected CIMT studies than as a meatanalysis. I propose to discuss the presented studies in more detail, please take into account the number of respondents, type of intervention, time of immobilization, time from stroke, location of stroke, clinical evaluation of the respondents.

Author Response

Response to reviewer comments

Reviewer 3

Sl.no

Queries

Response to queries

Line numbers

1

Thank you for the opportunity to review this interesting manuscript.

The point was: The effect of CIMT on balance has not been fully explored, and the existence of ambiguity in this treatment technique requires further investigation. Therefore, the aim of the current study was to evaluate the influence of CIMT on balance and functional mobility in patients after a stroke.

The study was planned correctly and in accordance with the accepted standards.

Thank you so much for your valuable time to review and for considering our study. The suggestions offered by you were immensely helpful. We carefully considered changes in the manuscript as per your suggestions. We modified and highlighted the changes in the manuscript. The highlighted changes are in yellow color. Here is a point-by-point explanation of reviewer comments. Thank you once again

N/A

2

In addition:

I suggest that each article should be cross-referenced in Tables 1 and 2. This will make it easier for the reader to find a suitable article in the references.

Thank you so much for pointing out. As per your suggestion we have provided references for the studies in the tables. We completely agree that this will enhance the comfort of the reader.

Table 1: 151 -152

Table 2: 162 - 167

3

The presented study presents a differentiated CIMT methodology in terms of the type of restriction, duration of treatment and type of intervention in the control group.

One study used additional NDT therapy? Did this intervention not affect the results of this study?

Only one study used an additional factor, a shoe insert

One study looked at gait with restricted upper limb

One study does not look at people after stroke.

Only 3 patients participated in one study.

Moreover, some of the subjects were in the subacute phase, others in the chronic phase.

We carefully considered your comments and reported impact of interventions and chronicity in the discussion.

252 – 266

307 – 312

323 – 325

330 -339.

4

I find the manuscript interesting in general, but more as an overview of selected CIMT studies than as a meta-analysis. I propose to discuss the presented studies in more detail, please take into account the number of respondents, type of intervention, time of immobilization, time from stroke, location of stroke, clinical evaluation of the respondents.

We carefully considered your comments and discussed various factors affecting the results of meta-analysis.

252 – 266

307 – 312

323 – 325

330 -339.

Round 2

Reviewer 1 Report

 The effort made by the authors has methodologically improved the study. It would be necessary to include, in Table 1, the number of studies found in each database, whose sum would give the total number that is expressed in the flow chart.

Author Response

Response to reviewer comments

Thank you so much for your valuable time to review and for considering our study. The suggestions offered by the reviewers were immensely helpful. We carefully considered changes in the manuscript as per your suggestions. We modified and highlighted the changes in the manuscript. The highlighted changes are in yellow color. Here is a point-by-point explanation of reviewer comments. Thank you once again.

Reviewer 1

Sl.no

Queries

Response to queries

Line Numbers

1

The effort made by the authors has methodologically improved the study. It would be necessary to include, in Table 1, the number of studies found in each database, whose sum would give the total number that is expressed in the flow chart.

In table 1 the necessary changes are made.

Table 1

Reviewer 3 Report

Thank you, I am satisfied.

Author Response

Response to reviewer 3 comments

Reviewer 3: Thank you, I am satisfied.

Answer:  Thank you so much for your valuable time to review and for considering our study. The suggestions offered by you were immensely helpful.
